# Recurrence and Eigenfunction Methods for Non-Trivial Models of Discrete Binary Choice

**DOI:** 10.3390/e25070996

**Published:** 2023-06-29

**Authors:** James Holehouse

**Affiliations:** The Santa Fe Institute, 1399 Hyde Park Road, Santa Fe, NM 87501, USA; jamesholehouse@santafe.edu

**Keywords:** social choice, relaxation times, non-equilibrium, special functions

## Abstract

Understanding how systems relax to equilibrium is a core theme of statistical physics, especially in economics, where systems are known to be subject to extrinsic noise not included in simple agent-based models. In models of binary choice—ones not much more complicated than Kirman’s model of ant recruitment—such relaxation dynamics become difficult to determine analytically and require solving a three-term recurrence relation in the eigendecomposition of the stochastic process. In this paper, we derive a concise closed-form solution to this linear three-term recurrence relation. Its solution has traditionally relied on cumbersome continued fractions, and we instead employ a linear algebraic approach that leverages the properties of lower-triangular and tridiagonal matrices to express the terms in the recurrence relation using a finite set of orthogonal polynomials. We pay special attention to the power series coefficients of Heun functions, which are also important in fields such as quantum mechanics and general relativity, as well as the binary choice models studied here. We then apply the solution to find equations describing the relaxation to steady-state behavior in social choice models through eigendecomposition. This application showcases the potential of our solution as an off-the-shelf solution to the recurrence that has not previously been reported, allowing for the easy identification of the eigenspectra of one-dimensional, one-step, continuous-time Markov processes.

## 1. Introduction

Models of social choice have been a popular application of physics to economics for many decades. Among the literature of voter models [1], where physical models are interpreted in the light of economic agents to give qualitative predictions, one of the most interesting contributions was made by Kirman [2]. In [2], the Moran process [3] is reinterpreted for ants choosing between two food sources, with an explicit analogy to systems of agents deciding between two economic decisions. From an economic perspective, the interesting finding is that agents can coalesce on a single choice due to endogenous forces alone, a feature that is not present in deterministic models from general equilibrium theory [4,5].

Such simple models as Kirman’s model of ant recruitment have utility, but in reality different economic decisions do not have the same rates of recruitment or the same random rates of agents switching to a specific choice [6]. In short, some decisions are more appealing when people conform to them, and others can be more capricious. The decision rules of agents may also be dependent on more than one other agent, which can lead to very different dynamics, e.g., in the hypothetical case of a system of vacillating voters [7]. Although Kirman’s model of ant recruitment has been solved explicitly in time [6,8], solutions to more generalized versions of the model have been difficult to obtain. To see why, consider the master equation for a generalized binary decision process with a fixed number of agents *N*,
(1)$$\begin{array}{c} \partial_{t}P\left(n,t\right)=A\cdot P\left(n,t\right), \end{array}$$
which describes the time evolution of the probability of having *n* agents deciding on one decision (with N−n deciding the opposite) at a time *t*, given the dynamics between the agents specified in the (N+1)×(N+1) matrix A. Where only a single agent can switch decision at any one time, the matrix A is tridiagonal. Formally, the solution can be found via the matrix exponential [9], which subscribes to the following decomposition:(2)$$\begin{array}{c} P\left(n,t\right)=\sum_{i=0}^{N}C_{i}\left(n\right)e^{\lambda_{i}t}, \end{array}$$
where $\lambda_{i}\leq 0$ are the eigenvalues of A and $C_{i}\left(n\right)$ are the corresponding eigenvectors (with the eigenvalue $\lambda_{0}=0$ corresponding to the steady-state eigenvector). The *relaxation times* to equilibrium are then clearly $1/\lambda_{i}$ (as can be seen from dimensional analysis), with $1/\lambda_{1}$ being the dominant timescale where $\lambda_{1}$ is the non-zero eigenvalue with the smallest magnitude. In the case of Kirman’s standard model, the *leading relaxation behavior is independent of the number of agents [6], and depends only on the rate of random switching between the two decisions* [6,10]. Where the behaviors between agents occupying different decisions become asymmetric, both $C_{i}\left(n\right)$ and $\lambda_{i}$ become insoluble due to the three-term recurrence relation that defines them. In such cases, it has been shown that the fully asymmetric ant recruitment model admits Heun function eigenvectors [6] whose series coefficients satisfy a three-term recurrence, and whose eigenvalues satisfy a N+1 order polynomial. The nature of this three-term recurrence in models of social choice is the focus of our study, i.e.,
(3)$$\begin{array}{c} R_{j}C_{j+1}-{\tilde{Q}}_{j}C_{j}+P_{j}C_{j-1}=0,\;j\in \left\{0,1,2,\ldots \right\}, \end{array}$$
with the boundary conditions $C_{0}=1$ and $C_{-1}=0$, and where the coefficients $R_{j}$, ${\tilde{Q}}_{j}$ and $P_{j}$ have a general dependence on *j*.

Although commonly stated as being unsolved (e.g., see the section on Heun functions in the handbook of Maple [11]), some papers in the past decade have made progress in solving three-term recurrence relations. Recent work by Choun [12], from a series of studies that include [13,14], tackles the problem of solving the three-term recurrence relation defining the Heun function and looks to find the conditions under which Heun functions reduce to finite polynomials. Unfortunately, the proposed solution is difficult to verify and unwieldy (see Equation (Equation 5) [12]). Similar conclusions can be made regarding another solution to the general three-term recurrence relation by Gonoskov [15], wherein the author defines and utilizes *recursive sum theory* (see Equations (47) and (48) [15]). However, other approaches with greater applicability are found in the seminal work of Risken et al. [16,17], who study generalized recurrence relations, often with applications to Fokker–Planck equations, and solve them using continued fractions. The work of Haag et al. takes a similar approach [18], and it is shown how exact solutions to the one-dimensional master equation can be found in terms of continued fractions (a work that precedes the cited work of Risken).

In this paper, we solve a general three-term recurrence relation using a simple linear algebraic method reliant on analytic results from the inversion of tridiagonal matrices [19]. This leads to expressions for the sequence $C_{j}$ in terms of determinants of the tridiagonal matrix, which can be conveniently expressed in terms of the products of orthogonal polynomials. These expressions allow one to see the analytic structure of the $C_{j}$ in terms of well-known mathematical operations, and to easily derive the solutions of the three-term recurrence for the generating function describing the eigenvectors and eigenvalues of generalized models of discrete binary choice.

An application of recurrence relations of particular importance can be found in providing closed-form expressions for the Frobenius solutions of higher-order functions, whose coefficients in a series expansion are described by three-term (or higher-order) recurrence relations. These higher-order functions have been shown to be particularly relevant in the solutions of the master equations describing non-trivial models of binary choice [6] and community assembly [8]. By *higher-order*, we mean that the number of singularities defining the function is greater than the number defining the hypergeometric differential equation (i.e., more than three), in which case the coefficients in the series expansion satisfy a two-term recurrence relation and can be solved by either Pochhammer or gamma functions [Chapter 15] [20]. The next highest order Fuchsian differential equation with four regular singularities defines the *general Heun function*, whose Frobenius solutions satisfy a three-term recurrence relation. Due to the increasing complexity of problems considered in the physics literature, Heun functions are becoming increasingly common and describe solutions to problems in quantum mechanics [21,22,23] and general relativity [21,24], and have some applications to stochastic processes [25] (see the review of Hortaccsu [26] and the references therein for further examples). A closed-form derivation of the series coefficients in the Frobenius solutions of Heun functions would allow researchers to easily obtain expressions defining polynomial solutions to Heun’s differential equation, and in the process determine the relaxation spectra of non-trivial social choice models.

In Section 2, we solve the recurrence relation in Equation (Equation 3) using linear algebraic methods, leading to the main result of our paper, as given by Equation (Equation 15), and we relate this solution to the previous work on the solution of the recurrence via continued fractions [17] in Section 3. Then, in Section 4, we review both Heun’s general and confluent differential equations, and show how our general solution solves for the coefficients in the Frobenius solutions in closed form. In Section 5, we return to the master equations described in this introduction and show how our results in Section 2, Section 3 and Section 4 allow one to determine the relaxation rates to equilibrium in two models of social choice that have eigenfunctions described by functions for which the Frobenius solutions satisfy the three-term recurrence relation. Finally, in Section 6 we conclude the study.

## 2. Closed-Form Solution of a Three-Term Recurrence Relation

We begin by re-writing the three-term recurrence relation in Equation (Equation 3) as a matrix equation. First, we define
(4)$$\begin{array}{c} A=\left(\begin{array}{ccccc} R_{0} & \; & & \; & \\ -{\tilde{Q}}_{1} & R_{1} & \; & & \\ P_{2} & -{\tilde{Q}}_{2} & R_{2} & \\ 0 & P_{3} & -{\tilde{Q}}_{3} & R_{3} & \\ \vdots & \; & & \; & \ddots \end{array}\right) \end{array}$$
where A is a square infinite-dimensional lower-triangular matrix. Note that because A is lower-triangular, the eigenvalues of A are $R_{i}$ for i∈{0,1,2,…}. Then, the recurrence relation in Equation (Equation 3) is equivalent to the following:(5)$$\begin{array}{c} A\cdot \vec{C}=\vec{l}, \end{array}$$
where we have defined the infinite-dimensional column vectors,
(6)$$\begin{array}{c} \vec{C}=\left(\begin{array}{c} C_{1}\\ C_{2}\\ C_{3}\\ C_{4}\\ \vdots \end{array}\right),\;\vec{l}=\left(\begin{array}{c} {\tilde{Q}}_{0}\\ -P_{1}\\ 0\\ 0\\ \vdots \end{array}\right), \end{array}$$
where the only two non-zero elements of $\vec{l}$ are $l_{1}$ and $l_{2}$. Then, $\vec{C}$ is given by
(7)$$\begin{array}{c} \vec{C}=A^{-1}\cdot \vec{l}. \end{array}$$
Therefore, if we can find the inverse of A then we have solved for the general three-term recurrence relation in Equation (Equation 3). In the following, we denote the inverse elements of A as $\theta_{i,j}\equiv {\left[A^{-1}\right]}_{i,j}$, and carrying out the multiplication in Equation (Equation 7), we find
(8)$$\begin{array}{c} C_{i}={\tilde{Q}}_{0}\theta_{i,1}-P_{1}\theta_{i,2}. \end{array}$$
Hence, there are two sets of inverse elements that we require, those in the first and second columns of $A^{-1}$. To find the matrix inverse, we make use of Cramer’s rule [27],
(9)$$\begin{array}{c} \theta_{i,j}=\frac{{\left(-1\right)}^{i+j}M_{j,i}}{\det(A)}, \end{array}$$
where $M_{j,i}$ is a minor of A, i.e., the determinant of A with row *j* and column *i* removed, and the determinant of A is simply the product of the eigenvalues of A that is given by
(10)$$\begin{array}{c} \det\left(A\right)=\prod_{i=0}^{\infty }R_{i}. \end{array}$$
Although the determinant of an infinite matrix is not formally well-defined, we show in the Appendix A that one does not have to evaluate this infinite product as cancellation occurs with the minors $M_{j,i}$ in the numerator of $\theta_{i,j}$. The remaining task is then to find the minors $M_{1,i}$ and $M_{2,i}$. In Appendix A and Appendix B, we explicitly find expressions for $M_{1,i}$ and $M_{2,i}$ in terms of a sequence of polynomials in *x*,
(11)$$\begin{array}{cc} \phi_{0}^{i}\left(x\right) & =1,\;\phi_{1}^{i}\left(x\right)=x+{\tilde{Q}}_{i},\; \end{array}$$
(12)$$\begin{array}{cc} \phi_{j}^{i}\left(x\right) & =\left({\tilde{Q}}_{i-(j-1)}+x\right)\phi_{j-1}^{i}\left(x\right)-R_{i-(j-1)}P_{i-(j-2)}\phi_{j-2}^{i}\left(x\right),\;j\in \left\{2,3,\ldots ,i-1,i\right\}. \end{array}$$
where the superscript *i* (plus 1) gives the number of polynomials in the set, and the subscript *j* denotes the (j+1)-th recursively defined polynomial. This result was first shown by [19], and allows one to express the determinant of a tridiagonal matrix in an analytic and computationally convenient way. Note that the polynomials $\phi_{j}^{i}\left(x\right)$ are orthogonal following Shohat–Favard theorem [28], although we do not use this property directly. Then, we can express $\theta_{i,1}$ and $\theta_{i,2}$ as
(13)$$\begin{array}{cc} \theta_{i,1} & =\frac{\phi_{i-1}^{i-1}\left(0\right)}{\prod_{j=0}^{i-1}R_{j}}, \end{array}$$
(14)$$\begin{array}{cc} \theta_{i,2} & =\frac{\phi_{i-2}^{i-1}\left(0\right)}{\prod_{j=1}^{i-1}R_{j}}. \end{array}$$
This gives us our closed-form expression for $C_{i}$ in terms of the recursively defined orthogonal polynomials $\phi_{j}^{i}\left(x\right)$,
(15)$$\begin{array}{c} C_{i}=\frac{{\tilde{Q}}_{0}\phi_{i-1}^{i-1}\left(0\right)-R_{0}P_{1}\phi_{i-2}^{i-1}\left(0\right)}{\prod_{j=0}^{i-1}R_{j}}, \end{array}$$
which is the main result of the paper. It elucidates the functional dependence of $C_{i}$ on the coefficients $P_{j},Q_{j}$ and $R_{j}$ in the recurrence relation itself. Note that this result is much more compact, and its derivation much easier, than other solutions to three-term recurrence relations described in [12,15]. It also avoid restrictions on the $C_{i}$, such as those imposed in the solution of Risken [17], wherein the authors require that for some *N* it is large enough that $C_{N+1}=0$. We will see in the final section of the paper that this result allows us to easily find the conditions under which polynomial solutions to Heun’s differential equations, or indeed any function defined by a three-term recurrence relation, occur. However, first, we establish the connection between Equation (Equation 15) and the work of Risken et al. [16,17].

## 3. Relationship to Continued Fractions

As stated in the introduction, the most useful previous solutions to the three-term recurrence come in the form of continued fractions, which are more cumbersome than the results derived in the previous section. Here, we show the relationship between our solution and that of [16], showing how the coefficients $C_{i}$ are equivalently given by a finite product over a set of continued fractions.

In [16], Risken and Vollmer study the non-stationary recurrence relations of the scalar and vector type. The scalar type is defined by
(16)$$\begin{array}{c} \frac{dC_{j}\left(t\right)}{dt}=R_{j}C_{j+1}\left(t\right)-{\tilde{Q}}_{j}C_{j}\left(t\right)+P_{j}C_{j-1}\left(t\right),\;j\in \left\{0,1,2,\ldots \right\}, \end{array}$$
where for some finite value of *N*, $C_{N+1}\left(t\right)=0$, which can either be an artificial truncation leading to approximate $C_{j}\left(t\right)$, or exact in some special cases. Such time-dependent recurrence relations are common in the study of one-dimensional master equations and first-passage time processes [29,30,31,32,33]. To solve Equation (Equation 16), one can treat it as an initial value problem (as was also carried out in [18]), but the easiest way to solve it is as an eigenvalue problem. To achieve this, one makes the separation ansatz $C_{j}\left(t\right)={\tilde{C}}_{j}e^{-\lambda t}$, which leads to a homogeneous three-term recurrence relation of the type seen in Equation (Equation 3), explicitly,
(17)$$\begin{array}{c} R_{j}{\tilde{C}}_{j+1}-\left({\tilde{Q}}_{j}-\lambda \right){\tilde{C}}_{j}+P_{j}{\tilde{C}}_{j-1}=0. \end{array}$$
The problem of finding $C_{j}\left(t\right)$ is then split into two. First, one needs to find the expression governing the ${\tilde{C}}_{j}$. Second, one must find the eigenvalue λ. Finding λ is a classic problem of linear algebra, and amounts to finding the values of λ under which the following holds:(18)$$\begin{array}{c} \left|\begin{array}{ccccccc} \left(\lambda -{\tilde{Q}}_{0}\right) & R_{0} & \; & & \; & \\ P_{1} & \left(\lambda -{\tilde{Q}}_{1}\right) & R_{1} & \; & & \\ \; & P_{2} & \left(\lambda -{\tilde{Q}}_{2}\right) & \; & & \; & \\ \; & \; & & \ddots & \; & & \\ \; & \; & & \; & \left(\lambda -{\tilde{Q}}_{N-1}\right) & R_{N-1} & \\ \; & \; & & \; & P_{N} & \left(\lambda -{\tilde{Q}}_{N}\right) \end{array}\right|=0, \end{array}$$
for which there will be N+1 solutions for λ where ${\tilde{C}}_{N+1}=0$. Generally, this can be achieved numerically. Clearly, usage of the exponential ansatz reduces the problem of calculating the ${\tilde{C}}_{j}$ to the same problem as we initially consider in Equation (Equation 3), but Risken et al. [16,17] solve it using continued fraction methods, as opposed to the method of orthogonal polynomials introduced above. Consider the transformation $S_{j}={\tilde{C}}_{j+1}/{\tilde{C}}_{j}$, which transforms Equation (Equation 17) into
(19)$$\begin{array}{c} R_{j}S_{j}+\left(\lambda -{\tilde{Q}}_{j}\right)+\frac{P_{j}}{S_{j-1}}=0, \end{array}$$
which can be solved for $S_{j}$ to give the recursive relationship
(20)$$\begin{array}{c} S_{j}=\frac{-P_{j+1}}{\left(\lambda -{\tilde{Q}}_{j+1}\right)+R_{j+1}S_{j+1}}. \end{array}$$
This relationship can be iterated to give an expression for $S_{j}$ in terms of a continued fraction,
(21)$$\begin{array}{c} S_{j}=\frac{-P_{j+1}}{\lambda -{\tilde{Q}}_{j+1}-}\frac{R_{j+1}P_{j+2}}{\lambda -{\tilde{Q}}_{j+2}-}\ldots \frac{R_{N-1}P_{N}}{\lambda -{\tilde{Q}}_{N}},\;j\in \left\{1,2,\ldots ,N-2\right\}, \end{array}$$
with $S_{N}=0$ and $S_{N-1}=-P_{N}/\left(\lambda -{\tilde{Q}}_{N}\right)$. One can then recover the ${\tilde{C}}_{j}$ by noticing that
(22)$$\begin{array}{c} {\tilde{C}}_{j}={\tilde{C}}_{0}\prod_{i=0}^{j-1}S_{i}. \end{array}$$
From here, it is clear that Equation (Equation 15) is equivalent to a finite product over a set of continued fractions. The benefit of the result in Equation (Equation 15) is that it is valid even for non-physical recurrence relations that grow unboundedly, and there is no restriction that $C_{N+1}\left(t\right)=0$ for some values of *N*. The results of Risken et al. for scalar three-term recurrence can hence be seen as a special case of Equation (Equation 15). Of course, for many physical applications the recurrence does not grow unboundedly, as often the recurrence variable represents physical variables or probabilities. This, however, is not a restriction on the special functions considered in the next section.

## 4. Heun Functions

Heun functions have had increased popularity in the study of Markov processes as researchers attempt to make their models more general, and eigenfunctions of the master equation or Fokker–Planck equations they consider can no longer be described by hypergeometric or lesser-order functions. They are the solution to a differential equation with four regular singularities, given by the ODE [34,35],
(23)$$\begin{array}{c} \frac{d^{2}y}{dz^{2}}+\left(\frac{\gamma }{z}+\frac{\delta }{z-1}+\frac{\epsilon }{z-a}\right)\frac{dy}{dz}+\frac{\alpha \beta z-q}{z(z-1)(z-a)}y=0, \end{array}$$
whose singularities are at z=0,1,a and *∞*, around each of which one can form a Frobenius solution of two linearly independent general Heun functions. To ensure that the Frobenius indices at z=∞ are {α,β}, the relation α+β+1=γ+δ+ϵ must be satisfied. This ODE is known as *Heun’s general equation*, and it is the natural extension of the hypergeometric differential equation ([20], Section 15.2), being a second-order linear Fuchsian equation with four singularities. For clarity, we show the associated radii of convergence of each Frobenius solution of the general Heun equation in Figure 1. Confluent forms of the Heun function also arise through limits of the solution to Heun’s general equation. For example, merging the singularities of the general Heun function at z=a and z=∞, by taking a→∞ and simultaneously q,αβ,ϵ→∞ in such a way that $\epsilon /a\to -\epsilon^{\prime }$, $q/a\to q^{\prime }$ and $\alpha \beta /a\to \alpha^{\prime }$, one arrives at the confluent Heun equation,
(24)$$\begin{array}{c} \frac{d^{2}y}{dz^{2}}+\left(\epsilon^{\prime }+\frac{\gamma }{z}+\frac{\delta }{z-1}\right)\frac{dy}{dz}+\frac{\alpha^{\prime }z-q^{\prime }}{z(z-1)}y=0, \end{array}$$
which has two regular singularities at z=0,1 and an irregular singularity of rank 1 at z=∞.

In what follows, we simply relabel the parameters of the confluent Heun function by dropping the prime, i.e., $\epsilon^{\prime }\to \epsilon ,\alpha^{\prime }\to \alpha$ and $q^{\prime }\to q$. One can then merge the singularities in the confluent Heun equation to derive further confluent types of Heun functions ([20], Section 31.12). The Frobenius indices for each of Equations (Equation 23) and (Equation 24) around each regular singularity are well reported [20,34], and here we consider series solutions to Equations (Equation 23) and (Equation 24) with the Frobenius index of 0 (note that our results below can be trivially applied to the solution defined by the second Frobenius exponent at each singularity, but with the re-defined parameters $P_{j}$, $Q_{j}$ and $R_{j}$). This means assuming that y(z) around z=0 has the form
(25)$$\begin{array}{c} y\left(z\right)=\sum_{j=0}^{\infty }C_{j}z^{j}. \end{array}$$
Substituting this into Equations (Equation 23) and (Equation 24) results in the three-term recurrence relation defining the solution at z=0 with the Frobenius index of 0,
(26)$$\begin{array}{c} R_{j}C_{j+1}-\left(Q_{j}+q\right)C_{j}+P_{j}C_{j-1}=0,\;j\in \left\{0,1,2,\ldots \right\}, \end{array}$$
with the boundary conditions $C_{0}=1$ and $C_{-1}=0$. The coefficients in the recurrence relation are different for the general and confluent Heun equations, and can be derived through the standard substitution of Equation (Equation 25) into the respective Heun differential equation. For the general Heun equation in Equation (Equation 23), we have
$$\begin{array}{cc} P_{j} & =(j-1+\alpha )(j-1+\beta ),\\ Q_{j} & =j\left((j-1+\gamma )(1+a)+a\delta +\epsilon \right),\\ R_{j} & =a(j+1)(j+\gamma ), \end{array}$$
whereas for the confluent Heun equation, we have
$$\begin{array}{cc} P_{j} & =(1-j)\epsilon -\alpha ,\\ Q_{j} & =j(1-j)+j(\epsilon -\gamma -\delta ),\\ R_{j} & =(j+1)(j+\gamma ). \end{array}$$

Clearly, for functions of the Heun class, Equation (Equation 26) is essentially Equation (Equation 3) but with ${\tilde{Q}}_{j}=Q_{j}+q$, meaning that the $C_{j}$ can be solved directly by a slight modification to Equation (Equation 15),
(27)$$\begin{array}{c} C_{i}=\frac{q\phi_{i-1}^{i-1}\left(q\right)-R_{0}P_{1}\phi_{i-2}^{i-1}\left(q\right)}{\prod_{k=0}^{i-1}R_{k}}, \end{array}$$
where the orthogonal polynomials are now evaluated at the accessory parameter x=q. This result shows explicitly why we must have γ∉{0,−1,−2,…}, since this would lead to a zero in the denominator of $C_{i}$. Note that in cases where the parameters of the Heun functions are such that they reduce to polynomials, or to solutions valid at more than one singularity, the $C_{i}$ will consist of a convergent series as i→∞, even outside the standard radius of convergence [34]. However, in general, the results of Risken do not apply for general Heun or confluent Heun functions outside the standard radius of convergence [16].

## 5. Application to Relaxation Times in Models of Social Choice

In this section, we apply the above analytics to explore the relaxation times to equilibrium in two distinct models of social choice, modeled as continuous-time Markov processes. Using the generating function approach to the master equation, one can show that the eigenspectra that define the time-dependence in the dynamics can be found through imposing physical restrictions on the generating function, which accounts for the finite size of the agent populations. This also allows one to connect continued fractions to their equivalent polynomial expressions that define the eigenspectra. The key references for the examples in this section are [2,6,7].

### 5.1. Fully Asymmetric Binary Choice Model

Many social choices are well described by binary choice situations wherein a fixed number of *N* agents decide between a *left* choice and a *right* choice with respect to two influences, (1) a random switching of decision of each agent, and (2) a recruitment whereby agents deciding one way can recruit others to the same decision. Such models have become increasingly common as they are much more analytically tractable than multiple-choice scenarios [37], and many social decisions can be approximated as being for-or-against a specific choice (even in a multiple-choice scenario). This situation describes the model of ant recruitment used by Kirman [2] to show how endogenous interactions can induce polarity in the collective decisions of agents and that polarity does not necessarily require an exogenous force. The same model has been used in other contexts to describe genetic drift [3] and the dynamics of migration [8,38]. In simpler cases where the effects of recruitment are symmetric in both decisions, the binary choice model has been solved [6,8]. However, making the effects of recruitment asymmetric leads to non-trivial relaxation rates and eigenfunctions for the stochastic process [6]. The fully asymmetric system defining the binary choice model is given by
(28)$$L\overset{\left(N-n\right)\varepsilon_{1}+n\left(N-n\right)\mu_{1}}{\underset{n\varepsilon_{2}+n\left(N-n\right)\mu_{2}}{⇌}}R,$$
where the expressions above and below the arrows indicate the propensity (per agent per unit of time) for a reaction to occur (determined from mass-action kinetics [39]), *L* and *R* denote an agent deciding left or right, respectively, and it is assumed that each agent is equally likely to interact with any other, meaning network effects can be ignored [8]. $\varepsilon_{1}$ and $\varepsilon_{2}$ are the random switching rates from left-to-right and right-to-left, respectively, and $\mu_{1}$ and $\mu_{2}$ are the respective rates of recruitment. In the propensities, *n* denotes the number of agents deciding *right*, meaning that (N−n) agents decide *left*. Note that this is a second-order reaction scheme due to the interactions between *L* and *R* agents.

This reaction scheme corresponds to the master equation,
(29)$$\begin{array}{cc} \partial_{t}P\left(n,t\right)= & \left[\left(N-\left(n-1\right)\right)\varepsilon_{1}+\mu_{1}\left(n-1\right)\left(N-\left(n-1\right)\right)\right]P\left(n-1,t\right)\\ & +\left[\left(n+1\right)\varepsilon_{2}+\mu_{2}\left(n+1\right)\left(N-\left(n+1\right)\right)\right]P\left(n+1,t\right)\\ \; & -\left[\left(N-n\right)\varepsilon_{1}+n\varepsilon_{2}+\left(\mu_{1}+\mu_{2}\right)n\left(N-n\right)\right]P\left(n,t\right), \end{array}$$
where P(n,t) is the probability of observing *n* right-deciding agents at a time *t*. The standard next step is to introduce the generating function $G\left(z,t\right)=\sum_{n}P\left(n,t\right)z^{n}$ which converts the master Equation (a set of coupled first-order ODEs) into a single PDE, which we give in Section C.1. Using separation of variables one can show that $G\left(z,t\right)∼f_{\lambda }\left(z\right)e^{-\lambda t}$, and the PDE defining G(z,t) reduces to a second-order ODE in $f_{\lambda }\left(z\right)$ whose solution is a general Heun function (also see [6]),
(30)$$\begin{array}{c} f_{\lambda }\left(z\right)=H\left(a,q\left(\lambda \right);\alpha ,\beta ,\gamma ,0;z\right), \end{array}$$
where we have defined,
(31)$$\begin{array}{cc} a & =\mu_{2}/\mu_{1},\\ q(\lambda ) & =\frac{\left(\lambda -N\varepsilon_{1}\right)\left(N-1\right)}{\mu_{1}},\\ \alpha & =-N,\\ \beta & =\frac{\left(N-1\right)\varepsilon_{1}}{\mu_{1}},\\ \gamma & =-\left(N-1\right)\left(1+\frac{\varepsilon_{2}}{\mu_{2}}\right), \end{array}$$
where the parameters have the same meaning as introduced for the general Heun function in Section 4. Note that δ=0. In order for the $f_{\lambda }\left(z\right)$ to be physical, we require that the λ be chosen such that $f_{\lambda }\left(z\right)$ is a polynomial of order *N* in *z*. This amounts to choosing λ such that P(N+1,t)=0, for which we can easily find a polynomial defining this from Equation (Equation 27),
(32)$$\begin{array}{c} P\left(N+1,t\right)\propto q\left(\lambda \right)\phi_{N}^{N}\left(q\left(\lambda \right)\right)-R_{0}P_{1}\phi_{N-1}^{N}\left(q\left(\lambda \right)\right)=0, \end{array}$$
which is a polynomial in λ of order N+1, whose roots define the eigenspectrum of relaxation to the equilibrium state. The most salient aspect of this equation is that *the eigenvalues, and hence the relaxation timescales, no longer have a trivial dependence on the random switching and recruitment rates or even on the system size*. Therefore, the key results relating to the relaxation timescales for Kirman’s symmetric ant recruitment model, i.e., the leading relaxation behavior being dependent only on the random switching rate and relaxation being independent of the size of the system [6,10], no longer hold. One can show the equivalence between the finite continued fractions and the polynomials defining the eigenspectrum, where using the formula in terms of continued fractions in ([6], Equation (Equation 28)) and equating the q(λ) terms in each expression, one finds
(33)$$\begin{array}{c} \frac{1}{Q_{1}+q-}\frac{R_{1}P_{2}}{Q_{2}+q-}\cdots \frac{R_{N-1}P_{N}}{Q_{N}+q}=\frac{\phi_{N-1}^{N}\left(q\right)}{\phi_{N}^{N}\left(q\right)}. \end{array}$$
This allows one to easily find the rational fraction corresponding to a continued fraction of the above form in terms of orthogonal polynomials in *q*. This holds for any $P_{j},Q_{j}$ or $R_{j}$ for which there is some $C_{N+1}=0$.

### 5.2. The Vacillating Voter Model

We can also use our method to easily derive polynomials describing the eigenspectra of models whose eigenfunctions satisfy generating function ODEs that are more complex than Heun functions, as long as the special functions defining them have series expansions whose coefficients are described by a three-term recurrence. For example, consider the following third-order reaction scheme that describes so-called vacillating voters [7],
(34)$$L\overset{p_{d}\left(N-n\right)+\left(1-p_{d}\right)\frac{\left(N-n\right)n}{N-1}\left(1+\frac{N-n}{N-1}\right)}{\underset{p_{d}n+\left(1-p_{d}\right)\frac{\left(N-n\right)n}{N-1}\left(1+\frac{n}{N-1}\right)}{⇌}}R,$$
where *L* and *R* again correspond to two different decisions, but now with different dynamical rules as compared to the asymmetric binary choice model. The model was solved semi-analytically in [6]. The rules described by this process are as follows. An agent is chosen at random from the population, and with probability $p_{d}$ changes their decision. However, with probability $\left(1-p_{d}\right)$ the agent looks at the decision of another agent. If this agent agrees with the originally chosen agent nothing happens, but if there is a disagreement the original agent will then select another agent at random and perform the same procedure again. Only if both other agents selected by the original agent disagree with their current view will the original agent change their mind. As one might expect, this leads to quite different behaviors from the original voter model [40], including transient and steady-state trimodality [6,7].

Again, one can construct a master equation describing the dynamics of the vacillating voters and can write the corresponding generating function equation. In Section C.2 we show this, and again use the separation of variables to define the equation which $f_{\lambda }\left(z\right)$ satisfies, which is a third-order ODE in $f_{\lambda }\left(z\right)$ and the unspecified spectral parameter λ. Employing a series solution about z=0, i.e., $f_{\lambda }\left(z\right)=\sum_{j}C_{j}z^{j}$, one then finds the following recursion relation for the coefficients $C_{j}$:(35)$$\begin{array}{cc} \; & C_{0}=1,\;\left(N-1\right)\left(\left(N-1\right)p_{d}+N\left(1-p_{d}\right)\right)C_{1}-q\left(\lambda \right)C_{0}=0,\\ \; & R_{j}C_{j+1}-\left(Q_{j}+q\left(\lambda \right)\right)C_{j}+P_{j}C_{j-1}=0, \end{array}$$
with the condition that $C_{N+1}=0$, and where we re-define
(36)$$\begin{array}{cc} q(\lambda )= & \left(N-1\right)\left(p_{d}N-\lambda \right),\\ R_{j}= & \left(j+1\right)(j\left(-j^{2}+j-2\right)\left(1-p_{d}\right)\\ \; & +\left(1-p_{d}\right)N\left(N-1\right)+{\left(N-1\right)}^{2}p_{d}),\\ Q_{j}= & -j\left(1-p_{d}\right)\left(3N-2\right)\left(N-j\right),\\ P_{j}= & \left(j-1\right)\left(1-p_{d}\right)\left(j-2N\right)\left(j-N-1\right)\\ \; & -\left(N-1\right)p_{d}\left(\left(j-2\right)N-j+1\right), \end{array}$$
which has been taken directly from [6]. Again, the polynomial describing the eigenspectra will be given by Equation (Equation 32), but now with the redefined $q\left(\lambda \right),R_{j},Q_{j}$ and $P_{j}$.

## 6. Discussion

In this paper, we provided a closed-form solution to a general three-term recurrence relation that determines the relaxation spectra in non-trivial models of binary choice. This allowed us to express the sequence defined by the recurrence in terms of orthogonal polynomials that allow one to easily see the analytic structure of terms in the sequence. Our solution is not reliant on the convergence of the recurrence, unlike that of the continued fraction solution, meaning that it can be applied even in situations where the sequence defined by the recurrence grows unboundedly. We then showed how this result provides the series coefficients in the Frobenius expansions of Heun functions. In the final section, we used these analytics for Heun functions, and other special functions whose Frobenius solutions satisfy a three-term recurrence, to derive concise polynomial expressions that define the eigenspectra for relaxation to the steady state in two distinct models of social choice.

Our result has clear analytic use, e.g., easily computing the polynomial satisfied by the eigenspectrum of a continuous-time Markov process (Section 5) or expressing finite continued fractions as a rational fraction (Equation (Equation 33)). However, a computational limitation of our solution is that each $C_{j}$ will take the same order of time to compute a direct forward substitution on the triangular matrix equation in Equation (Equation 5), although solving via this method does not lead to a closed-form solution (i.e., each $C_{j}$ would depend on all $C_{i<j}$ preceding it). We note that the same restriction applies to the continued fraction solution to the three-term recurrence relation provided by Risken [16]. However, *it is often the analytical structure that is of interest to us in solving physical problems*—as we have shown in Section 5.

Several avenues for further study remain open. The first is the extension of the results presented herein to higher-order recurrence relations. Such an approach has been previously considered by Risken [16], wherein higher-order recurrence relations are converted into three-term *vector recurrence relations* that can be solved by continued fraction methods very similar to those used for three-term scalar recurrence relations. Using a similar approach, it may be possible to generalize the results in this paper to higher-order recurrence relations in a way that does not require the usage of matrix continued fractions. The results that we have presented also allow for connections to be drawn to other parts of the Markov process literature involved in solving one-dimensional master equations for various problems, such as its time-dependent solution with arbitrary rates, or the one-dimensional first-passage time probabilities with absorbing [32] and reflecting [41] boundaries [42]. These papers highlight the utility of studying the three-term recurrence under different boundary and initial conditions, and the results that we have found possibly allow for a unification of the results found therein. Finally, and most optimistically, it may be possible to use our methods to derive time-dependent solutions to chemical reaction networks involving reactions of bimolecular form, and multi-step reactions, and provide an extension to the generalized solutions of monomolecular reaction systems provided by Jahnke et al. in [43] and the solution to the one-dimensional, linear, one-step master equation [31]. The calculation of these results would rely on finding an appropriate representation of reaction schemes involving bimolecular reactions, possibly in the form of a vector recurrence relation, that then allows for linear algebraic methods to become computationally useful.

## Figures and Tables

**Figure 1 entropy-25-00996-f001:**
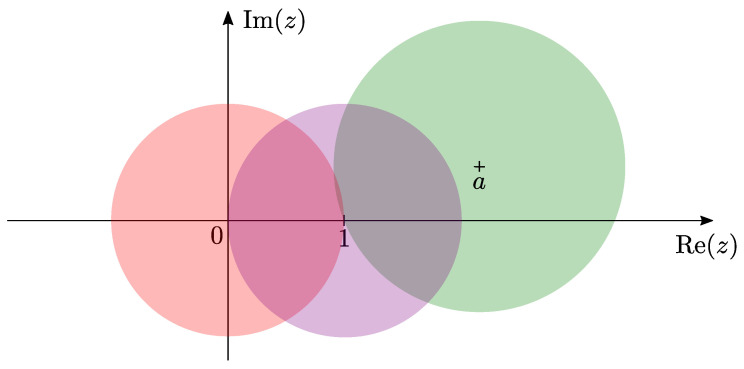
Illustration showing the radii of convergence of the Frobenius solutions of the general Heun equation about z=0,1 and *a*. The parameter *a* can take any complex value. Each Frobenius solution is valid in a circle in the complex plane (centered at the respective singularity) until the next singularity. Ordinary series expansions, not about singularities, are also valid in a circle extending up to the next singularity. Note that in cases where the Heun functions simplify to polynomials, or even Frobenius solutions valid at two singularities, the radii of convergence will extend beyond those in the illustration. The radius of convergence of the solution at z=∞ can be seen clearly through the independent variable transformation z→1/x, as shown in ([34], p. 15). For further details, see [36].

## Data Availability

No new data were created in this study.

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
