# Peer review of "Recurrence and Eigenfunction Methods for Non-Trivial Models of Discrete Binary Choice"

_entropy, 2023, doi:10.3390/e25070996_

Round 1
Reviewer 1 Report
In this manuscript, the author studies a different method to obtain solutions of three-term recurrence relations and thereafter compares it to previous methods. He shows different situations where this type of equations can be found, and how the method applies to those cases. Finally, he explores the applications to compute relaxation times in binary-state stochastic models.
The manuscript is very well written and properly contextualized within the literature, the methods are explained with clarity and its applicability is straightforward from reading the manuscript. For these reasons, I believe that it is acceptable for publication in Entropy as it is. From the point of view of results it may be lacking, but from the methodological perspective it is valuable.
Author Response
We thank the reviewer for the kind comments about the relevance of our study. Please find the updated version of the manuscript attached, with the changed sections in blue.

Reviewer 2 Report
The author obtains the solution of a three term recursion and establishes its relationship to the the solution obtained by continued fractions and to the general and confluent Heun functions. The author then discusses two problems from the literature to demonstrate the methodology.
I found the paper engaging, it is well written, well paced and goes to the point. For a general reader, like myself, it is easy to find the main results and follow the logic that leads to them without going through the proofs. But I have some critical remarks. Neither the title nor the abstract accurately represent the work. The title implies a study of relaxation in discrete binary choice systems but the work is a mathematical derivation of closed form solutions. Only section V is devoted to binary choices, and even then as a means of demonstrating the method, not as a study of how relaxation occurs. And while the abstract begins with the discussion of binary models, the Introduction is about recurrences, and so are the next three sections. This is disorienting to the reader until section V, where the examples are finally discussed. The author has two options, either focus on the derivations or on the models that they solve. For the readership of Entropy I recommend the latter. I believe minor editing ought to fix this. The introduction, just as the abstract, should start with the models. And Section V, which introduces the models rather quickly and without enough detail, should be expanded. If the author can say something about "relaxation," that would be great, otherwise the term should be removed from the title.
Overall this is an interesting paper and should be published after revision.
Author Response
We thank the reviewer for finding our study engaging and for the well-founded feedback on the motivation behind the paper. To address concerns raised by the reviewer, namely that the introduction and abstract are misrepresentative of the contents of the study, we have completely written the introduction to place the emphasis on models of social choice. Additionally, some standard methods have been included in the introduction to emphasize the problems we are attempting to solve in the paper. The abstract and title have also been re-written in this vein and are now more representative of the content of the study. What is meant by relaxation has been clarified in the introduction and now elaborated on in Section V. Please find the updated version of the manuscript attached, with the changed sections in blue.
